# Evaluation of Kitchen Waste Recycling as Organic N-Fertiliser for Sustainable Agriculture under Cool and Warm Seasons



Ksawery Kuligowski [1,*] , Izabela Konkol [1] , Lesław Świerczek [1] , Katarzyna Chojnacka [2] , Adam Cenian [1] and Szymon Szufa [3]

1. Physical Aspects of Ecoenergy Department, The Institute of Fluid-Flow Machinery, Polish Academy of Sciences, Fiszera 14 St., 80-231 Gdansk, Poland
2. Department of Advanced Material Technology, Faculty of Chemistry, Wroclaw University of Science and Technology, M. Smoluchowskiego 25 St., 50-372 Wroclaw, Poland
3. Faculty of Process and Environmental Engineering, Lodz University of Technology, Wolczanska 213 St., 90-001 Lodz, Poland
* Correspondence: kkuligowski@imp.gda.pl

**Abstract:** Kitchen waste could be processed and recycled into safe fertilizers/soil improvers for sustainable agriculture through different methods: (1) Dried pellets from model kitchen waste treated with anaerobic effective microorganisms; and (2) Anaerobically digested kitchen waste. For comparison, a commercial mineral fertilizer was used. These methods were applied in two separate glasshouse experiments: one under cool (mainly winter) conditions (X–IV) and one under warm (mainly summer) conditions (VI–X) consisting of 3–4 subsequent harvests in northern Poland. Comparing the food waste agronomic performance after anaerobic digestion and effective microorganism treatments, especially under different climatic conditions, is a novel approach. Kitchen waste served as a much better fertilizer than mineral fertilizer, but only during the cool season. In addition, it provided 20–40% more plant yields for dosages >120 kg N/ha and a similar N uptake. In the warm season, in comparison to effective microorganism-incubated kitchen waste, its anaerobic digestion improved the relative agronomic effectiveness twice after 30 days of growth (82% versus 43%). However, the total effectiveness for anaerobically digested kitchen waste versus pelleted and effective microorganism-incubated kitchen waste was 32% versus 27% (N utilization-wise) and 36% versus 21% (plant biomass yield-wise). The Monod kinetic model was applied for the internal efficiency of N utilization; for the best fitting procedure, $R^2 > 0.96$ for the cool season and $R^2 > 0.92$ for the warm season. Kitchen waste introduced to the soil provided better soil properties than mineral fertilizer. The study contributes to the biological systems for waste recycling in agriculture, bioproduction processes, and the global food chain.

**Keywords:** food waste; kitchen waste; nitrogen uptake; ryegrass growth; agronomic effectiveness; Monod kinetics



## 1. Introduction

In simple terms, recycling consists of the reprocessing of waste in industrial processes in order to obtain substances or materials for primary or other purposes. Thanks to recycling, we can recover raw materials from packaging that became waste after use, for example. The lack of raw materials in the future may pose a serious threat to many countries. The EU Framework Directive 2008/98/EC obliges member states to take all necessary measures to build a European resource-efficient recycling economy in order to minimize this threat to future generations in Europe.

The traditional economy, based on the use of raw materials from non-renewable sources, generates significant amounts of hazardous waste and mismanagement, which leads to environmental problems [1]. Recently, new concepts of waste-free production and recycling and technologies based on alternative sources of raw materials have been

developed. These solutions are closely related to the circular economy and green/cleaner production and recycling. The aforementioned systems have been developed in response to environmental problems, decreasing non-renewable material resources and promoting a transformation from an economy based on linear material flows to one based on circular material flows [2].

Recycling is a modern solution to the problem of waste. By replacing natural resources with secondary ones, we implement the assumptions of a resource-efficient low-emission economy, supporting sustainable development.

The annual production of municipal waste in Poland amounts to 12,500 mt, of which around 3269 mt is recycled (26% on average), including around 1012 mt that is digested and composted. In 2018, a typical Pole produced 325 kg of waste/year (<200 kg/year for rural communes and up to 384 kg in urban ones); 38% of municipalities produced less than 200 kg of municipal waste per inhabitant (mainly rural communes), including two communes below 50 kg. In 53% of municipalities, the amount of generated waste was in the range of 200–400 kg per inhabitant. The largest amounts of municipal waste are generated in touristic municipalities; in six of them, >1000 kg of domestic waste per inhabitant were collected. In comparison, the EU average is 480 kg per year per capita. The EU leader in waste production is Denmark at 780 kg, while Poland is the second-lowest producer after Romania (280 kg). There are 192 installations of mechanical and biological waste processing in Poland, and 195 installations for bio-waste composting. In 2018, there were 2144 publicly available points for selective collection of domestic waste, of which 37% was located in cities and 63% in rural areas. That same year, more than 3.6 mt (94 kg/inhabitant) were collected selectively, of which up to 3.3 mt were from households (29% of the total amount of municipal waste). In 2018, 28% of the total amount of domestic waste was biodegradable waste, which includes food waste of plant origin with high biogas potential that results in over 3.5 mt of biowaste in Poland in total. Even if it is composted or anaerobically digested, it is still available in the form of compost, or a digestate, for novel management techniques e.g., the fertilization of urban green areas [3].

The organic fraction of municipal solid waste (kitchen waste) separated at the source constitutes a vast quantity of valuable, clean, and nitrogen-rich products. The valorization of food waste is actually one of the important current research areas [4].

Different approaches, including prevention, mitigation, and post-valorization, might be proposed for food waste management. Waste utilization strategies can be divided into chemical or biochemical (composting, digestion, fermentation) and thermal (gasification, incineration, pyrolysis) [5–12].

Kitchen waste is defined as organic residues from home kitchens, but food (plate) waste (FW) is defined as food that has not been completely eaten, or spoilt food. Kitchen waste and food waste contain a high level of organic matter and fertilizer nutrients. For this reason, FW has potential economic value for the production of fertilizer [13].

Kitchen waste, apart from other factors (plant protection products, transport, urbanization), poses a serious threat to the emission of nutrients into the environment. For example, nitrogen emissions from food production and consumption generate significant environmental pressure. Food production depends on the use of fertilizers. Nutrients originating from the fertilizers will be found in food, where they can be recovered using various processes.

Biological methods for the processing of organic waste have a strong position in waste management. Both waste processing technologies, composting and fermentation, have advantages and disadvantages. The choice of composting or fermentation will depend on specific local conditions [14]. In recent years, kitchen waste has been used to produce second-generation biofuels, including ethanol [7,12].

Anaerobic digestion is an efficient method for kitchen waste treatment and disposal. Pre-treatments, co-digestion, the dosing of additives, and process optimization are effective measures to alleviate the inhibition of hazardous kitchen waste components on the performance of the anaerobic digestion process. The reuse of treated residue can significantly

increase the additional value of derived products from anaerobic digestion and improve the commercial value of kitchen waste for biogas projects [6], whereas composting is an aerobic process in which a succession of different microbial populations degrades the original organic substrates into a more physically and chemically stable product [15].

In the market, a few commercial microbial inoculants (improving composting process) are available. One of these is the effective microorganisms (EM) [11]. Effective microorganisms are a consortium of selected species of beneficial microorganisms, including five families, 10 genera, and more than 80 types of anaerobic and aerobic microbes, including lactic acid bacteria, photosynthetic bacteria, yeast, actinomycetes, and fungi [11,16]. Effective microbes are widely used in agriculture and can be applied as inoculants to increase microbial activities and diversity of agricultural waste. After the treatment of different types of organic waste by composting process, they can provide a cost-effective biological method. Composting with microorganisms enhanced the total microbial population and biodiversity. It also increased the rate of degradation, bulk density, and mineralization [11,17].

The literature positions cited below often do not distinguish which EM are used, aerobic or anaerobic, nor give exact commercial names of the substrates applied. This could question, for example, the use of anaerobic EM in aerobic treatments, such as composting. Zhong et al. [15] evaluated the composting process with and without EM. They observed the growth parameters of studied plants. The addition of EM during composting improved the properties of the compost, but in the case of plant growth, there were no significant differences. Jusoh et al. [17] observed a significant difference in the content of macro- and micro-elements. The use of EM increased the content of N and K in compost. However, general composting with or without EM follows a very similar course. Fan et al. evaluated the effect of EM on the home-scale composting of organic wastes. The properties and parameters of both composts (one with EM, another without) were comparable, with the difference being the smell of the compost improved significantly with the addition of EM [11]. Hu and Qi [18] conducted a long-field experiment to investigate the effect of the application of compost on crop growth and yield. They concluded that the long-term application of EM in combination with composting improved crop yields. The highest NPK content in plant tissues with EM compost treatment demonstrated a higher efficiency of the release of nutrients through organic and microbial application. The researchers deduced that the effect of improving soil fertility was better in compost with EM application than without [18].

Liu et al. [19] developed a slow-release phosphate fertilizer using coatings derived from waste cooking oil. Cooking oils were used as substrates to obtain polyurethane coatings, which were modified by multi-walled carbon nanotubes. The agricultural value and usefulness of kitchen waste oil were confirmed in the production of biodegradable coatings for controlled release fertilizer. In turn, Qi et al. [20] made polyvinyl alcohol-based urea coatings to obtain a fertilizer with a controlled release of nutrients.

Prado et al. [21] performed analyses for the valorization of kitchen waste in fertilizers. The following methods of valorization were proposed: co-digestion with black sewage and the composting process, as well as landfill disposal. Solid fertilizers were used, but the methodology was not indicated. A process of anaerobic co-digestion using kitchen waste and livestock manure [22–24] was also applied.

Liu et al. [25] used kitchen waste as one of the components of solid fertilizer. The fertilizer was obtained by combining straw biochar, which was modified with chitosan. The composition was supplemented with a mineral NPK fertilizer. The agronomic properties of the fertilizer were positively evaluated for soil deacidification and increased soil fertility; a low content of toxic elements was confirmed. Furthermore, this fertilizer contains chitosan, enabled $Pb^{2+}$ ions adsorption, which means that it could be used to reduce the bioavailability of toxic elements in contaminated soils. The bioavailability of Pb for plants decreased by 50%. Beneficial effects on soil health have been confirmed, including an increase in the content of organic matter and assimilable forms of nitrogen and phosphorus.

It also beneficially affects soil microorganisms, including their diversity. Increased nitrogen retention in the soil was also confirmed.

Xing et al. [26] developed a new approach to fertilizers from kitchen waste. The goal was to reduce nitrogen emissions by identifying the flow of raw materials and modeling the dynamics of the system. The authors stressed that the recovery of kitchen waste, supplemented by manure and straw, could lead to a 40% reduction in nitrogen emissions in food production. Attention was also drawn to the need to use integrated plant cultivation and animal husbandry in local systems to achieve circular agriculture.

Karwal and Kaushik [27] performed a biological transformation of kitchen waste enriched with lawn waste and buffalo manure in the vermicomposting process (*Eisenia foetida*). In this way, a fertilizer was obtained, which was a humus carrier using a combined method of composting and vermicomposting in a process lasting 3 months. The best feedstock ratio for kitchen waste (lawn waste: buffalo manure) was found to be 6:3:1.

The available scientific literature shows that fertilizer formulations are rarely limited to a single biological component. They are usually the co-digestion or co-composting of kitchen waste with other waste. After this initial processing, the biobased residues obtained can be used as one of the components for further formulation. Therefore, other additives (e.g., sorbents or agents supporting granulation) are introduced, and the composition is corrected with the use of mineral salts. In this way, it is possible to obtain an organic mineral fertilizer complying with regulations with a standardized content of fertilizer nutrients and pathogens as well.

Nutrient recovery and reuse practices have a potential to address the most pressing problems related to nutrient use in the food chain, such as pollution, depletion of finite resources (such as P), and waste management [28].

In general, the literature review of various waste-based fertilizers points to the fact of slower mobilization of nutrients from the organic fertilizer than from the mineral one and the right-shifted response of ryegrass-to-fertilizer curve for the organic one. This is due to the nutrient bounds in organic compounds, which was already stated in the previous author's project work [3]. However, some issues are still unknown, e.g.,

- The effect of winter conditions on plant growth for both fertilizers; which fertilizer will resist them better?
- The overfertilization effect—where and at what stage of growth does it occur?
- The nitrogen uptake inhibition effect—will it be dependent on fertilizer application rate or the level of impurities?

This experiment should provide the following results:

- Plant biomass increases as a function of the fertilizer load, i.e., g plant d.m. = f (kg N/ha),
- N uptake by the plant as a function of fertilizer load, i.e., g plant N/kg plant d.m. = f (kg N/ha applied),
- Internal N utilization expressed as an increase in plant biomass as a function of N uptake in kg plant d.m./ha = f (g plant N/ha raised),
- Soil residual properties such as Soil pH, Soil conductivity, Soil N content.

The agronomic properties of fertilizers are carried out in plant tests. The most common are germination tests in which the germination strength, germination index (percentage of seeds germinated), and phytotoxicity are determined. These tests allow the effect of fertilizers to be assessed during the initial phase of plant growth. Pot tests also allow subsequent phases (e.g., up to the third leaf in the case of cucumbers) to be tested under controlled conditions in the phytotron or greenhouse. The advantage of germination tests is that they are carried out under near-real conditions and allow the assessment of the so-called transfer factor, i.e., the uptake of nutrients in the soil-plant system. Various model plants are used in pot trials, including ryegrass [29–31], as in the present study.

This paper presents the effect of using (1) EM-incubated kitchen waste and (2) anaerobically digested kitchen waste as organic N-fertilizers in both cool (winter) and warm (summer) seasons under glasshouse conditions. The study evaluated the effect of kitchen

waste application on plant growth, biomass yield, and nitrogen uptake. The Monod model/kinetics is used to describe biomass growth; this is the model commonly used in natural, biological, ecological, and environmental studies [32,33].

In addition, this paper proposes a complete quantitative fertilization model based on literature review, which forms a closed-loop resource chain from the beginning to the end to achieve efficient management and full quantitative consumption of nutrients.

Since previous studies mainly focused on the effects of EM addition to the composting process, this paper describes another approach to valorizing kitchen waste without a composting step, using both effective microbe (EM) treatments and anaerobic digestion (AD) for the subsequent production of solid organic fertilizers (OF). The production of non-toxic organic fertilizers and using non-polluted natural resources are very important because this is the way to achieve a clean environment.

## 2. Materials and Methods

The effect of kitchen waste on the growth of ryegrass was studied in pot experiments carried out in two greenhouses: Glasshouse 1 (G1)—Prokowo, Pomerania, Northern Poland (cool season from X 2020 to IV 2021); and Glasshouse 2 (G2)—Gdynia Wiczlino, Pomerania, Northern Poland (warm season from VI 2022 to X 2022). In Glasshouse G1, experiments related to MF and KW applications were conducted during a six-month period (October 2020–April 2021), whereas in G2, the effects of fertilizers based on MF, KW, and KW-dig were investigated during a four-month period (July–October 2021).

### 2.1. Fertilisers

#### 2.1.1. Model Waste Preparation

In total, two experimental models, as well as processed kitchen waste, were investigated in two glasshouse experiments and compared with the commercially available mineral NP fertilizer. First, the model kitchen waste [34] has been prepared according to the following recipe: 25 g each of apple, lemon, roll, butter, sour cream, milk, cottage cheese, yoghurt, eggs, meat with bones, sausage, fish meat, potatoes, banana, tomato, lettuce, fruit juice, and bun, as well as 50 g of flowers and paper. These components were ground to a particle size of less than 5 mm and mixed well to obtain homogeneous mass.

#### 2.1.2. Kitchen Waste Conversion and Fertilizer Production

The obtained basic substrate was then processed into two fertilizers. KW fertilizer was obtained by the basic substrate inoculation with commercially available (mainly anaerobic) effective microorganisms (Greenland Technologia EM Ltd., Janowiec, Poland). Inoculation was achieved by dispersing 1 mL of the bacterial product in 250 mL of deionized water and mixed with 1 kg of substrate. The substrate was then collected in a sealed plastic container with a vent tube for two weeks. After fermentation process, the substrate was partly dried to about 70% dry mass and formed into pellet-shaped granules, then dried to obtain the stable mass (ca. 95% d.m.). KW–dig fertilizer consisted of residues after methane digestion of KW. Mesophilic methane digestion was carried out in 2-L reactors for 30 days in accordance with the methodology described by Konkol et al. [9]. The fermentation residue was centrifuged in a laboratory centrifuge (MPW 260RH) for 10 min at $5000 \times g$ RPM without the use of coagulants. The prepared KW–dig fertilizer was stored at 4 °C until used in the greenhouse tests. The composition of the applied EM is as follows: lactic acid bacteria, photosynthetic bacteria, yeast, nitrobacteria, cane molasses; total nitrogen (0.3%), $K_2O$ (0.2%). Table 1 shows the basic characteristics of the materials used.

#### 2.1.3. Reference Mineral Fertilizers

Two commercially available mineral fertilizers were used. One was dedicated for autumn (FLOROVIT NPK with additional Mg and Fe contents (cool season)) and one for spring (FLOROVIT NP "fast growth" with additional S and Fe contents (warm season)).

**Table 1.** Basic characteristics of fertilizer materials applied.

| Material | Season | Symbol | d.m. | TOS | N-Total | P-Olsen | P-Total | K-Olsen | K-Total |
|---|---|---|---|---|---|---|---|---|---|
| *Unit* | | | % | % | | | g/kg (Pure Ingredient) | | |
| *Soil* | Cool | | 84.75 | 5.80 | 1.26 | 0.0195 | 0.191 | 0.0670 | 0.626 |
| *Soil* | Warm | | 88.77 | 6.08 | 1.32 | 0.0190 | 0.186 | 0.0649 | 0.610 |
| *Kitchen Waste EM-incubated/Dried & Pelleted* | Cool, Warm | KW | 25.75/ 95.00 | 93.72 | 34.18 | 0.0486 | 1.511 | 5.715 | 8.482 |
| *Kitchen Waste digested* | Warm | KW-dig | 16.51 | 95.00 | 42.82 | 0.0486 | 1.511 | 5.715 | 8.482 |
| *Mineral fertiliser FLOROVIT NPK* [1] | Cool | MF | 100 | 0 | 44.00 | 14.8 | 22.7 [3] | 157.3 | NA |
| *Mineral fertiliser FLOROVIT NP* [2] | Warm | MF | 100 | 0 | 190 | 17.03 | 26.2 [3] | NA | NA |

[1] additionally 2.5% MgO and 4% Fe, [2] additionally 25% $SO_3$ and 6% Fe, [3] P soluble in neutral ammonium citrate.

### 2.1.4. Assuming Fertilizer Dosages

Presuming that the plant responds to rich-N organic waste, fertilizer dosages were applied at rates ranging from 20 to 270 kg N/ha (in the cool season) and 370 kg N/ha (in the warm season) to reach the plateau on the N response curve. Table 2 shows the experimental plan expressed by the fertilizer dose assumption and the amounts of the corresponding (calculated) nitrogen and fertilizer per pot. It was decided to start with the normal dose (20 kg N/ha) that is recommended for ryegrass according to the mineral fertilizer requirements. The dose was increased by 50 kg N/ha until it reached 170 kg N/ha, which is the maximum allowed N amount yearly for natural fertilizers on Polish agricultural land [35]. The dose was increased further; the last dose was 270 kg N/ha (for the cool season) and 370 kg N/ha (for the warm season, assuming higher growths) to reach the plateau of over-fertilization.

**Table 2.** Amounts of kitchen waste fertilizers based on N content, added to the soil in the glasshouse experiment.

| Mineral Fertiliser NPK (Cool Season)—MF (C) | | | | | Kitchen Waste Incubated with EM, Pelleted (Cool and Warm Seasons)—KW | | | | | |
|---|---|---|---|---|---|---|---|---|---|---|
| Dosage No | kg N/ha | g N/pot | g Fertiliser/Pot | mg N/kg Soil d.m. | Dosage No | kg N/ha | g N/Pot | g of Fertiliser/Pot *Cool/Warm* | | mg N/kg Soil d.m. |
| 1 (normal) | 20 | 0.033 | **0.75** | 0.024 | 1 | 20 | 0.033 | **0.97** | **1.02** | 0.023 |
| 2 | 70 | 0.116 | **2.63** | 0.083 | 2 | 70 | 0.116 | **3.38** | **3.56** | 0.079 |
| 3 | 120 | 0.198 | **4.50** | 0.142 | 3 | 120 | 0.198 | **5.79** | **6.10** | 0.135 |
| 4 (max in PL) | 170 | 0.281 | **6.38** | 0.201 | 4 | 170 | 0.281 | **8.21** | **8.64** | 0.192 |
| 5 | 220 | 0.363 | **8.25** | 0.260 | 5 | 220 | 0.363 | **10.62** | **11.18** | 0.248 |
| 6 | 270 | 0.446 | **10.13** | 0.319 | 6 | 270 | 0.446 | **13.04** | **13.72** | 0.304 |
| | | | | | 7 * | 370 | 0.611 | | **18.81** | 0.417 |
| Mineral fertiliser NP (warm season)—MF (W) | | | | | Kitchen Waste digested, (warm season)—KW–dig | | | | | |
| Dosage No | kg N/ha | g N/pot | g fertiliser/pot | mg N/kg soil d.m. | Dosage No | kg N/ha | g N/pot | g fertiliser/pot | | mg N/kg soil d.m. |
| 1 (normal) | 20 | 0.033 | **0.17** | 0.023 | 1 | 20 | 0.033 | **4.67** | | 0.023 |
| 2 | 70 | 0.116 | **0.61** | 0.079 | 2 | 70 | 0.116 | **16.34** | | 0.079 |
| 3 | 120 | 0.198 | **1.04** | 0.135 | 3 | 120 | 0.198 | **28.01** | | 0.135 |
| 4 (max in PL) | 170 | 0.281 | **1.48** | 0.192 | 4 | 170 | 0.281 | **39.69** | | 0.192 |
| 5 | 220 | 0.363 | **1.91** | 0.248 | 5 | 220 | 0.363 | **51.36** | | 0.248 |
| 6 | 270 | 0.446 | **2.35** | 0.304 | 6 | 270 | 0.446 | **63.03** | | 0.304 |
| 7 * | 370 | 0.611 | **3.21** | 0.417 | 7 * | 370 | 0.611 | **86.37** | | 0.417 |

* Dosage nr 7 applied only under warm season.

### 2.1.5. Fertilizer Application

The levels of N added are shown in Table 2.

### 2.2. Soil and Plants

### 2.2.1. Soil Preparation

The plants were grown in the < 2-mm sieved fraction of a sandy soil mixed with peat in *w/w* ratio sand/peat = 5/1, which corresponds to *v/v* ratio 1:1.5. Soil properties are

reported in Table 1. Moreover, the following parameters of soil were determined: pH 7.48, the redox potential 33.39 mV, electrical conductivity EC 191.6 μS/cm in cool season experiment, and pH 8.29, the redox potential 63.8 mV, and electrical conductivity EC 159.7 μS/cm in warm season test. Approximately 1.75 kg of prepared fresh soil were placed in a 14.5-cm internal diameter pot (surface area: 0.0165 m$^2$). Supplemental nutrient solutions (except N) were added to each pot according to the recipe: $K_2SO_4$ (42 g/L) 12 mL/pot and 6 mL/pot in solution $CaCl_2 \cdot 2H_2O$ (90 g/L), $MgSO_4 \cdot 7H_2O$ (24 g/L), $MnSO_4 \cdot H_2O$ (6 g/L), $ZnSO_4 \cdot 7H_2O$ (5.4 g/L), $CuSO_4 \cdot 5H_2O$ (1.2 g/L), $H_3BO_3$ (0.42 g/L), $CoSO_4 \cdot 7H_2O$ (0.16 g/L), $Na_2Mo_4 \cdot 2H_2O$ (0.12 g/L). The soil in each pot was then pre-watered with 120 mL of deionized water (DIW), then the soil and nutrients were thoroughly mixed in the top 5 cm of the soil.

### 2.2.2. Ryegrass Growth

Then, 80 grains of annual ryegrass, or 0.5 g (a mixture of *Lolium perenne* 40%, *Lolium multiflorum-estanzuela* 20%, *Festuca rubra* 25%, *Lolium hybridum* 15%), were placed on the surface of the soil and covered with an additional 80 g of soil. Experiments were carried out in duplicate, and pots were re-randomized every 7 days to eliminate differences in insolation and kept at constant weight with (DIW) at field capacity 20% (g $H_2O$/g soil d.m.), that is, approximately 26.4% (cm$^3$ $H_2O$/cm$^3$ soil). Harvesting was carried out every month over a 4-month period by cutting the tops approximately 1 cm above the soil surface. The harvested plants were then placed in paper bags and dried at 105 °C until the weight was constant.

### 2.3. Soil and Plant Analysis

Soil samples were analyzed for pH and EC (1:5 H2O), as well as for soil nitrogen (total), phosphorus (Olsen and total), and potassium (Olsen and total) before planting, and for pH and EC and total nitrogen after the last harvest [36]. The phosphorus concentration in liquid samples (soil extracts) was determined on a portable spectrophotometer (Hach DR3900, Hach Company, Ames, IA, USA) using the Hach Method 8048, with the mineralization step. Before analysis, water-soil samples were filtered on a paper filter, followed by a 0.45-μm syringe filter. The tops of the ryegrass after each of the four harvests were dried and ground. The samples were analyzed for Total Kjeldahl Nitrogen (TKN). The samples were digested (SpeedDigester K-436, Büchi Labortechnik AG, Flawil, Switzerland) in concentrated $H_2SO_4$ acid in the presence of a titanium-based catalyst. The next step was the steam distillation step (K-355 distillation unit, Büchi Labortechnik AG, Flawil, Switzerland) into boric acid solution with a Tashiro indicator, then titrated with HCl acid to measure the released ammonia.

### 3. Theory and Calculation

### 3.1. Agronomic Effectiveness

Absolute agronomic effectiveness (AAE) and relative agronomic effectiveness (RAE) of the materials were calculated for each of the four harvests and cumulatively after the end of the experiment using the N uptake data (1–4 harvests and the cumulatively) and dry matter yield data (only cumulatively). AAE is expressed by a slope of the best-fit line of the relation between plant N uptake and N application rate, and RAE is expressed by the ratio of AAE of the studied to AAE of the reference fertilizer [37]. This is a standard method to evaluate the performance of various fertilizers.

### 3.2. The Monod Model/Kinetics

The Monod model/kinetics was applied, and it is used to describe biomass growth. The model is commonly used in natural, biological, ecological, and environmental studies [32,33].

## 4. Results

*4.1. Measurements*

4.1.1. Response of the Ryegrass Biomass Yield to Kitchen Waste-Based Fertilizers

The dry matter yields versus the fertilizer dosage range constitute the first and fairly visible measure of fertilizer performance. Figure 1 shows the response of the dry matter yield to a given fertilizer application rate.

COOL SEASON (DAYS 30–180) data initially presented in internal project report [3].

In general, the findings after 180 days of growth support the hypothesis of slower mobilization of nutrients from the KW organic fertilizer and gradual growth of the plant over the next growth months (maximum after 30 and 90 days around 0.16 g/pot). It provided at least 30%, 60%, and 100% larger biomass yields after 30, 90, and 180 days, respectively, than MF at 170 kg N/ha and further with higher loads. The ultimate effect of using KW was at least 60% better than MF when discussing cumulatively for doses >170 kg N/ha. This was in contrast to mineral fertilizer, which allowed the ryegrass to grow quite quickly in the beginning but only up to 70 kg N/ha (30 and 90 days), after which the response curve slowly declined (the ryegrass did not further grow for doses > 70 kg N/ha). The exception was observed after 180 days, where the relative maximum growth for the MF was observed at 120 kg N/ha, perhaps due to the nitrogen accumulation in the soil over long-lasting frost that was observed in February and March. Summing up:

1.  Harvest 1 (30 days): Dynamic growth up to 0.18 g d.m./pot for 70 kg N/ha for the MF and gradual growth of ryegrass grown on the KW with rapid increase at 120 kg N/ha; both response curves cross at ca. 100 kg N/ha.
2.  Harvest 2 (next 60 days, 90 days in total): Opposite situation—gradual growth for the MF up to 0.13 g d.m./pot for 70 kg N/ha and dynamic growth of ryegrass grown on the KW up to 0.15 g d.m./pot (i.e., higher than the mineral one); the response curves cross at ca. 100 kg N/ha.
3.  Harvest 3 (next 90 days, 180 days in total): Dynamic growth up to 0.15 g d.m./pot for 120 kg N/ha for the MF and gradual growth of ryegrass for KW with rapid increase at 120 kg N/ha, but only up to 0.07 g d.m./pot at 170 kg N/ha; response curves cross at ca. 160 kg N/ha.
4.  Overfertilization for the MF: for all three harvests after 70 kg N/ha; for KW: NO observed.

Figure 1 (lowest to the left) shows the cumulative amounts of ryegrass harvested during the three harvests (180 days). Growths varied from 0.15 to 0.5 g d.m./pot for the MF and from 0.20 to 0.4 g dm/pot for the KW, with maximums reaching ca. 0.42 g d.m./pot at 70 kg N/ha for the MF and 0.38 g d.m./pot at 170 kg N/ha for the KW. The KW-based fertilizer has reached the response plateau for doses larger than 170 kg N/ha without a clear overfertilization effect, as growths were still moderately high (0.35–0.40 g d.m./pot). MF at these doses provided ca. 40–60% smaller biomass increments amounting 0.25 g d.m./pot for dose 170 kg N/ha and dropping rapidly to 0.15 g d.m./pot for each increasing dose. So, there is a larger buffer capacity in the case of KW to apply it with higher dosages without harming plants.

WARM SEASON (DAYS 30–120).

The first month resulted in growth in the range of 2–2.5 g d.m./pot for KW–dig (the highest among the studied materials). These are similar growths for MF, but only for a dose of at least 220 kg N/ha. Meanwhile, the lowest values (i.e., 1.5–2 g d.m./pot) were found for KW. This is probably due to the fact that the higher amounts of $NH_4NO_3^-$ contained in digested kitchen waste (KW–dig) are better available to plants.

This delivered ammonium nitrate is immediately available to plants, allowing just-in-time application. The remaining solvated ammonium hydroxide ($NH_4OH$) is nitrified by soil bacteria to form nitrates ($NO^{3-}$) and volatile nitrous oxide ($N_2O$), but with fewer emissions because the ammonium nitrate concentration is reduced after partially being taken up by plants. Excess nitrate will still be lost either by denitrification or by the leaching of the groundwater (in real-scale conditions, not in a glasshouse).

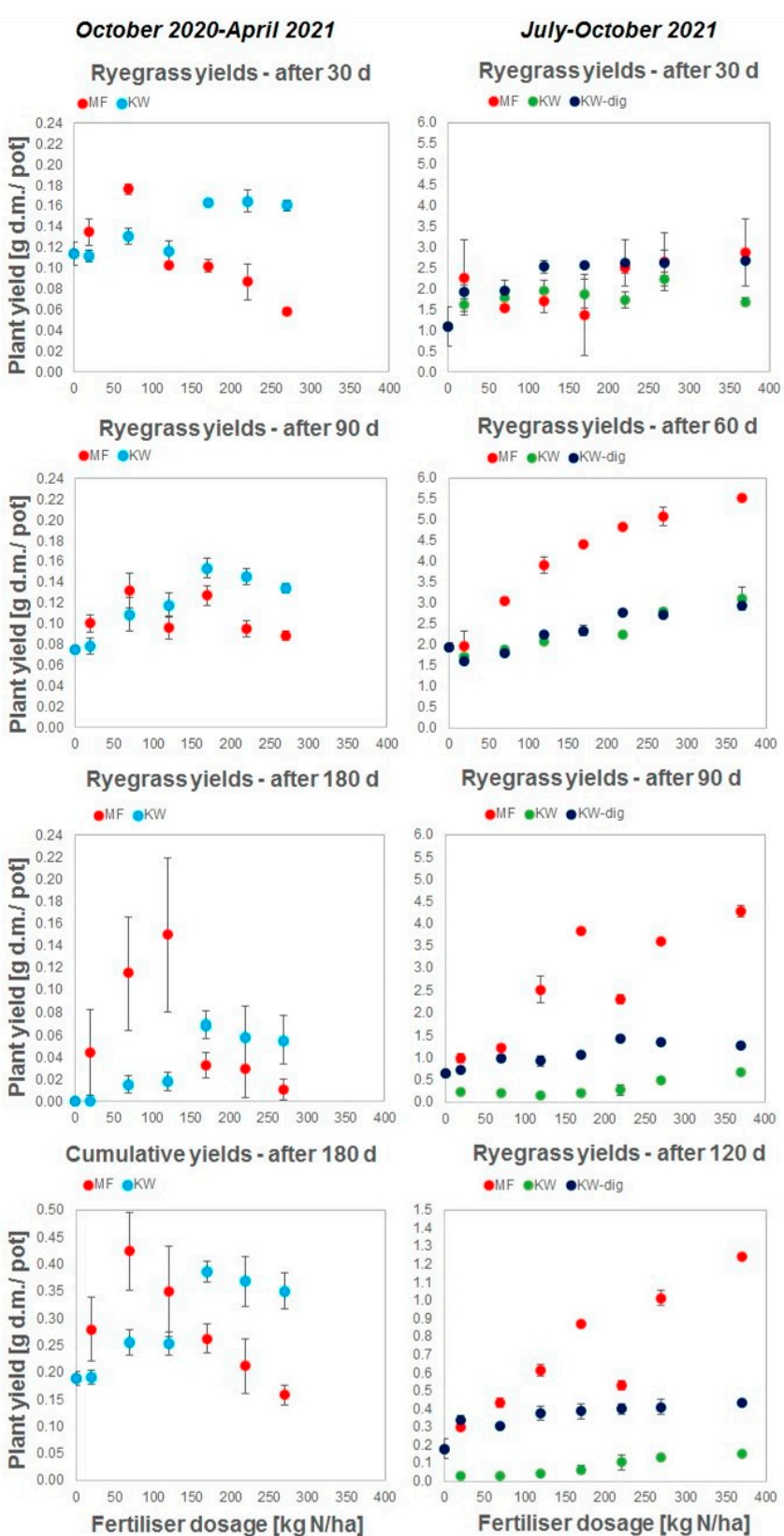

**Figure 1.** Ryegrass biomass yields response to kitchen waste-based fertilizers after subsequent harvests, compared to the mineral fertilizer (MF) for harvests after 30, 90, and 180 days; cumulative for cool season (**left—three harvests and cumulative**); and after 30, 60, 90, and 120 days for warm season (**right—four harvests**). Standard deviations included, insignificant if not visible.

The KW–dig fertilizer provided 65% better increments than MF for doses up to 170 kg N/ha. When the root network expanded in the second month, the grass grew better on MF (2.0–5.5 g d.m./pot), while the performances of KW and KW–dig were comparable (1.5–3 g d.m./pot). After 90 and 120 days, the growth response to MF was much better (up to 4 g d.m./pot after 90 days and up to 1.3 g d.m./pot after 120 days) than for KW (max 0.6 g d.m./pot after 90 days and max 0.15 g d.m./pot after 120 days) and KW–dig (max 1.5 g d.m./pot after 90 days and max 0.4 g d.m./pot after 120 days).

KW–dig provided around three and four times more ryegrass than KW after 90 and 120 days, respectively. The weaker growths during the last month (October) were also related to temperature and radiation declines.

### 4.1.2. Nitrogen Uptake by Ryegrass Fertilization with Kitchen Waste-Based Fertilizers

COOL SEASON (30–180 days).

Figure 2 shows that the N content in the grass samples varied in the range of 32 and 87 g N/kg d.m., and its concentration was increasing linearly with the increasing fertilizer application rate. The N content was quite stable in the range of 75 and 85 g N/kg d.m. after 30 days for both fertilizers and with a dose of 70 kg N/ha. After the next 60 days (90 in total), the N content reached a maximum MF application dose of 65 g N/kg d.m. at 120 kg N/ha, while for KW, the N uptake was similar only for a dose of 220 kg N/ha. After 180 days, KW provided better N uptake at a dose of 220 kg N/ha (up to 60 g N/kg d.m.) compared with MF (i.e., up to 50 g N/kg d.m.). This shift in better N uptake at higher doses for KW is probably related to more nitrogen delivered to the soil, but with less availability than from MF. However, by calculating the total N uptake per area and comparing it with the fertilizer application rate, it was concluded that both fertilizers provided equal N uptake (12 kg N/ha) at a 120 kg N/ha application rate (utilization of ca. 10% of N applied). KW had much better effects on the N uptake at higher doses (16 to 19 kg N/ha for the organic KW compared to 14-to-9 kg N/ha for MF).

WARM SEASON (DAYS 30–120 days).

The first month (June/July) provided ca. 10–20% better N uptake in plants fertilized with MF reaching 55 g N/kg dm at 170 kg N/ha dose. Later the growths were similar for KW–dig and MF, reaching 52 g N/kg d.m. at 220 kg N/ha. KW provided similar uptakes (50 g N/kg d.m.) but only at 270 kg N/ha and greater. This also shows the shift of response maximums to higher application dosages for organic fertilizers as compared with mineral ones, at the same time proving greater N availability from anaerobically digested materials (KW–dig) than from undigested ones (KW). The next months (i.e., after 60 days) provided an N uptake maximum of 30, 25, and 15 g N/kg d.m. for MF, KW, and KW–dig, respectively, and 22, 25, and 15 g N/kg d.m. after 90 days for MF, KW, and KW–dig, respectively. This trend shows that undigested KW leads to uptakes comparable to MF at later times, whereas digested KW–dig provided steady and lower uptakes over the wide range of doses. In the last month (after 120 days), KW again provided up to a 20% greater uptake at higher dosages (>170 kg N/ha) than MF and KW–dig (their uptakes were generally lower and oscillated for about 15–20 g N/kg d.m., with MF being slightly greater).

The N utilization calculated from both the N uptake and the ryegrass d.m. yield is presented in Figure 3 for all harvests. Linear relationships of N use by plants as a function of fertilizer dosage are shown for some fertilizers and harvests, where regression was possible. The slope of the fitted lines is a measure of absolute agronomic effectiveness (AAE).

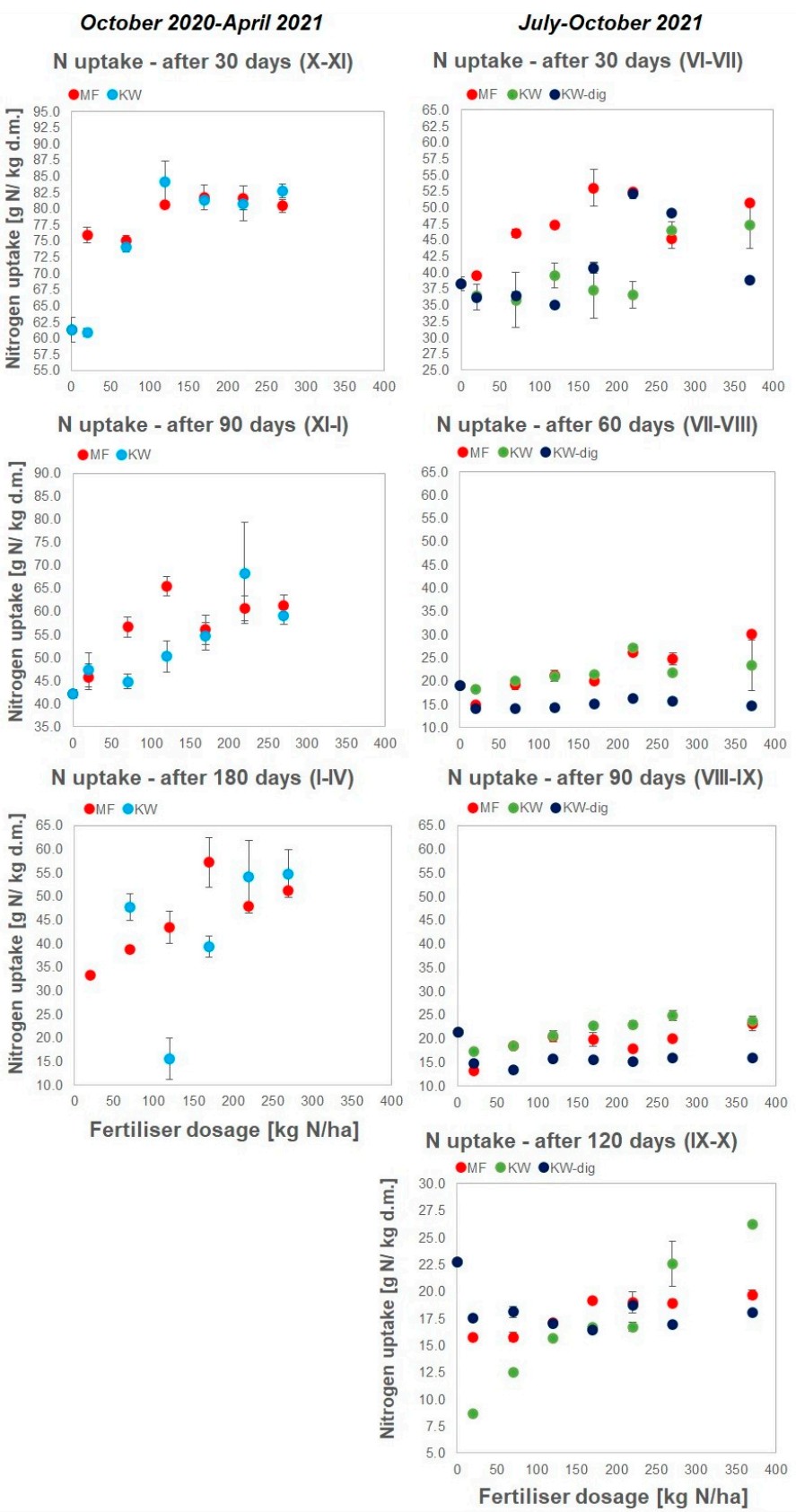

**Figure 2.** Nitrogen uptake by ryegrass fertilization with kitchen waste after subsequent harvests compared to mineral fertilizer (MF) for the cool season (**left—three harvests**) and for the warm season (**right—four harvests**). Standard deviations included, insignificant if not visible.

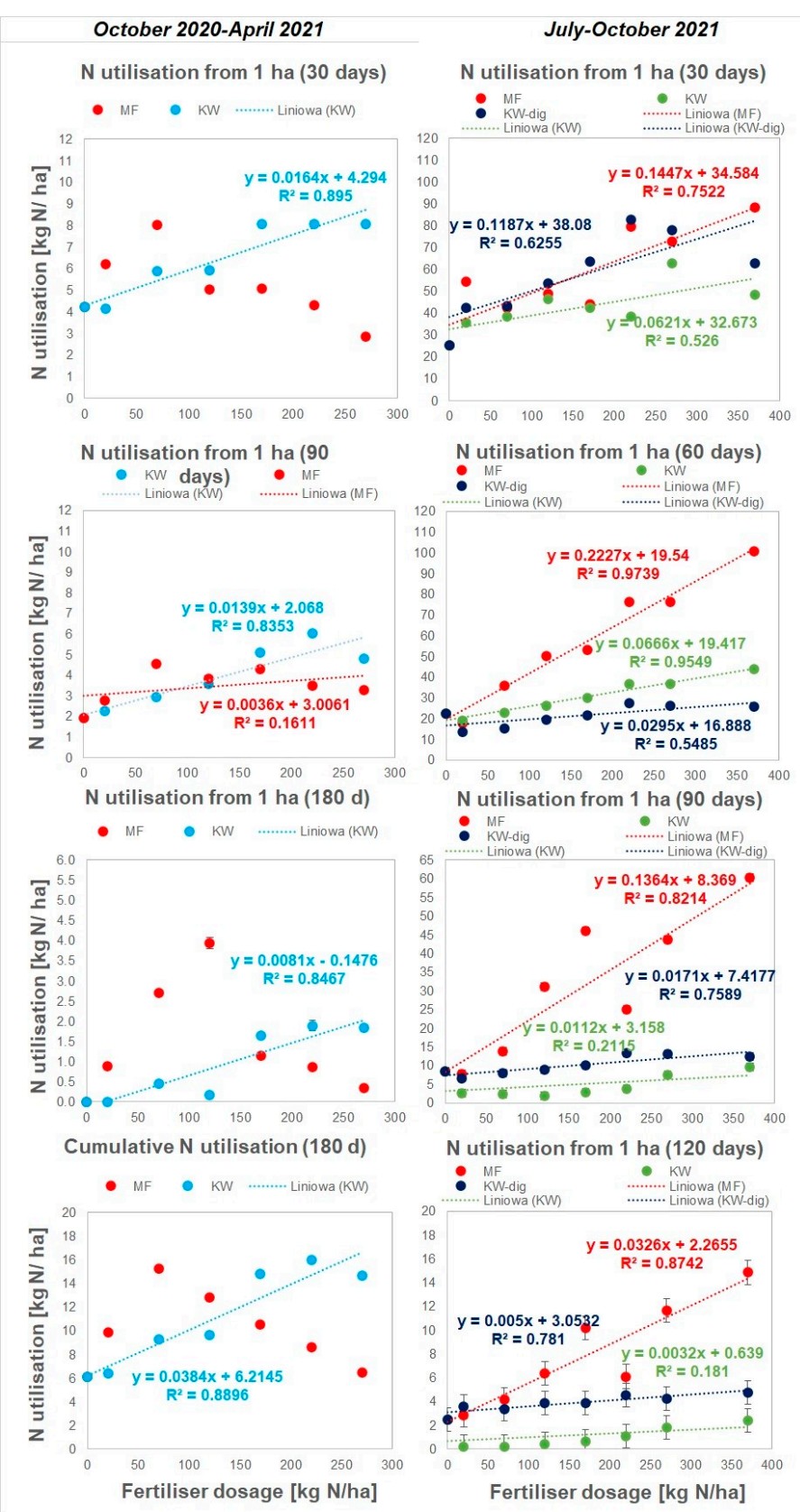

**Figure 3.** Efficiency of nitrogen use of 1 ha of ryegrass fertilization with kitchen waste (KW and KW–dig) related to mineral fertilizer (MF). Cool season data (**left**) and warm season data (**right**). Standard deviations included, insignificant if not visible.

In the cool season, total nitrogen use per ha, for doses greater than 120 kg N/ha, is always much greater after KW application as related to MF application. For high application rates, in kg N used/ha, KW allows at least 100% more N use (4 vs. 8) after 30 days, 20–100% more N use (3 vs. 6) after the next 60 days, at least 60% more N use (1 vs. 1.6) after the next 90 days, and, in total, 50–150% more N use (10 vs. 15) than MF. This supports the effects described in Figures 1 and 2. For the cool season, the calculation of AAE was possible for KW for 30 days, 90 days, 180 days, and a cumulative scenario with a moderate linear regression of the response to N use ($R^2 > 0.84$). None or weak linear regressions (after 90 days only) were found for the response to N and the use of MF because (as mentioned earlier in Figures 1 and 2) higher loads of this fertilizer resulted in over-fertilization and inhibition of the growth.

Under warm conditions, the situation is the opposite, meaning that MF always provided better N use than KW and KW–dig. The use of N from soils amended with KW–dig (30–80 kg N/ha) was comparable with the MF scenario (30–90 kg N/ha) only during the first 30 days, i.e., at the beginning of the test, while the KW scenario led to a lower use (30–60 kg N/ha). The following months of growth resulted in the increasing levels of N use with an increased MF fertilizer application rate: 20–100 kg N/ha, 10–60 kg N/ha, 2–15 kg N/ha for 60, 90, and 120 days, respectively. N use by plants in these months grown in the soil amended with KW–dig/KW were around 4/3 (after 60 days), 6/10 (after 90 days), and 4/7 (after 120 days) smaller than N use in the MF scenario. Apart from the second month of growth (60 days), the application of anaerobically digested KW–dig led to around a 50–150% greater N use than undigested KW, showing the highest differences in the first month (30 days) and fourth month (120 days). For the warm season calculation of AAE, moderate linear regression was possible for MF ($R^2 = 0.75$–0.97), while poor linear regression was possible for KW–dig (0.63–0.78, excluding growth at 60 days). A good linear regression for the response to N use in the application of KW was found only after 60 days ($R^2 = 0.95$).

### 4.1.3. Residual Soil Properties after the Growth of Ryegrass

COOL SEASON (30–180 days), data initially presented in internal project report [3]

After 180 days of testing, fertilizers did not provide any more significant growths of the ryegrass due to the intensive winter season (long-lasting frosts and limited sunlight). Thus, the experiment concluded and the residual effects on the soil were examined. Figure 4 shows that the residual nitrogen content in the soil was around 20–100% higher after KW application (0.7–1.3 g N/kg d.m.) than for MF (0.5–0.7 g N/kg d.m.), with a maximum for KW at a 220 kg N/ha fertilizer application rate. The more fertilizer that was applied, the more N that was left in the soil bank. A dynamic response increase was very noticeable for KW doses > than 120 kg N/ha, where soil-N remained stable; MF applications had to be very high (270 kg N/ha) to leave comparable amounts of N in the soil (<1 g N kg d.m.).

This again shows that the more N is applied with organic matter, the more it remains in the soil bank, whereas more nitrate from mineral fertilizer is lost via denitrification as $N_2O$ and/or is taken up by plants. This indicates that this substantial bank of probably organically bound remaining N in the soil still provides maximum growths at very high application rates for the organic KW (shown in Figure 1). The residual N in soils amended with mineral fertilizer is rather stable over the entire spectrum of fertilizer application rates, which is connected with its fast mineralization, uptake, and air emissions after denitrification to $N_2O$. This is not reflected in a better grass growth in the whole range of dosages.

The pH remains fairly stable (8.45–8.30) with an increased organic KW application rate, whereas it drops from 8.45 to 7.80 after the application of MF, slightly influencing the acidity of the soil. The electrical conductivity of soil increases from 245 to 420 μS/cm for the MF application, while the application of the KW provides a very insignificant drop of EC (from 120 to 80 μS/cm) due to organic compounds that are not leaching any significant

amounts of anions from the soil solution. So again, when KW is used, the soil remains unaffected by the increase in salinity.

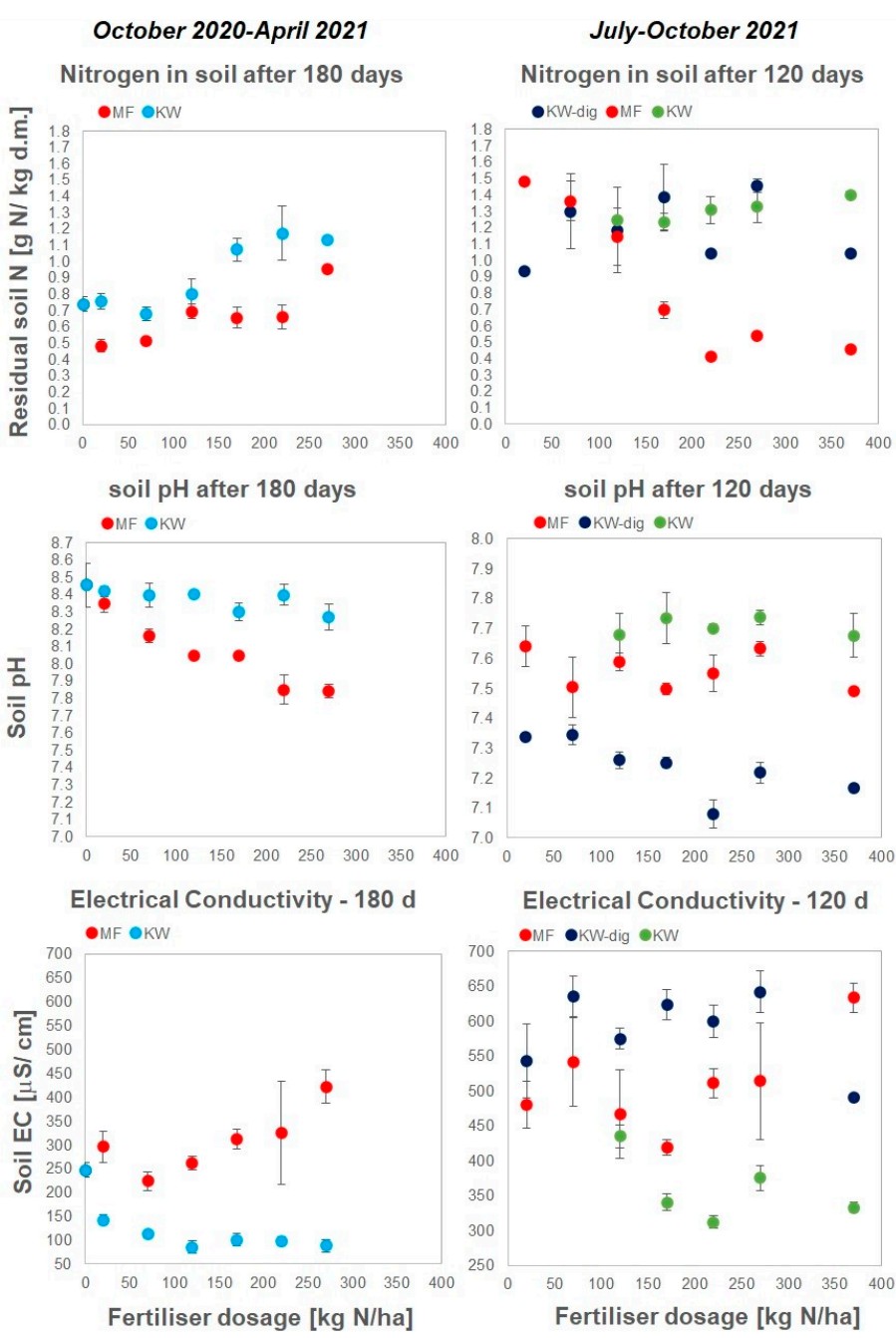

**Figure 4.** Residual soil properties after fertilization with kitchen waste (KW) after 180 days (cool season) and after 120 days (warm season) compared to mineral fertilizer (MF). Standard deviations included, insignificant if not visible.

WARM SEASON (DAYS 30–120).

The residual N content in the soil was generally higher for KW and KW–dig (from 0.9 to 1.4 g N/kg d.m.) than for MF (from 0.4 to 1.5 g N/kg d.m.) and even up to three times higher at 220 kg N/ha and more. The soil pH was quite stable across application rates and was on average 7.6–7.8 for KW, 7.5–7.6 for MF, and 7.1–7.3 for KW–dig. The EC varied after application of KW–dig from 550 to 650 μS/cm and increased after MF application from 400 to 600 μS/cm (especially after 220 kg N/ha and more), while it decreased from 450 to

300 μS/cm for KW, which was also observed in the cool season. Again, significant growths of EC were only observed for MF applications (Figure 4).

### 4.2. Modelling

The internal efficiency curves (Figure 5) show the d.m. yield as a function of plant N utilization for all fertilizers for each harvest in both seasons. The response curves were fitted using the Monod kinetics model (first and third columns), then the calculated and measured yields were compared (figures in the second and fourth columns). Additionally, a power model (fourth column—90 and 120 days) was related to the linear function of predicted versus measured data. The model is described by the following Equation (1):

$$Y = Y\text{max} \times \frac{Ni}{Km + Ni} \tag{1}$$

where $Y$ represents the yield of ryegrass (in t d.m./ha) as a function of $N$ (kg) used by plant; $Ymax$ is the maximum yield; and $Km$ is the $N$ at which $Y$ equals half of the maximum yield. The best-fit values of the $Y_{max}$ and $Km$ for each of the three harvests (cool season) or four harvests (warm season) are shown in the first and third columns in Figure 5, and the function relating the fitted v. measured values is plotted in the second and fourth columns.

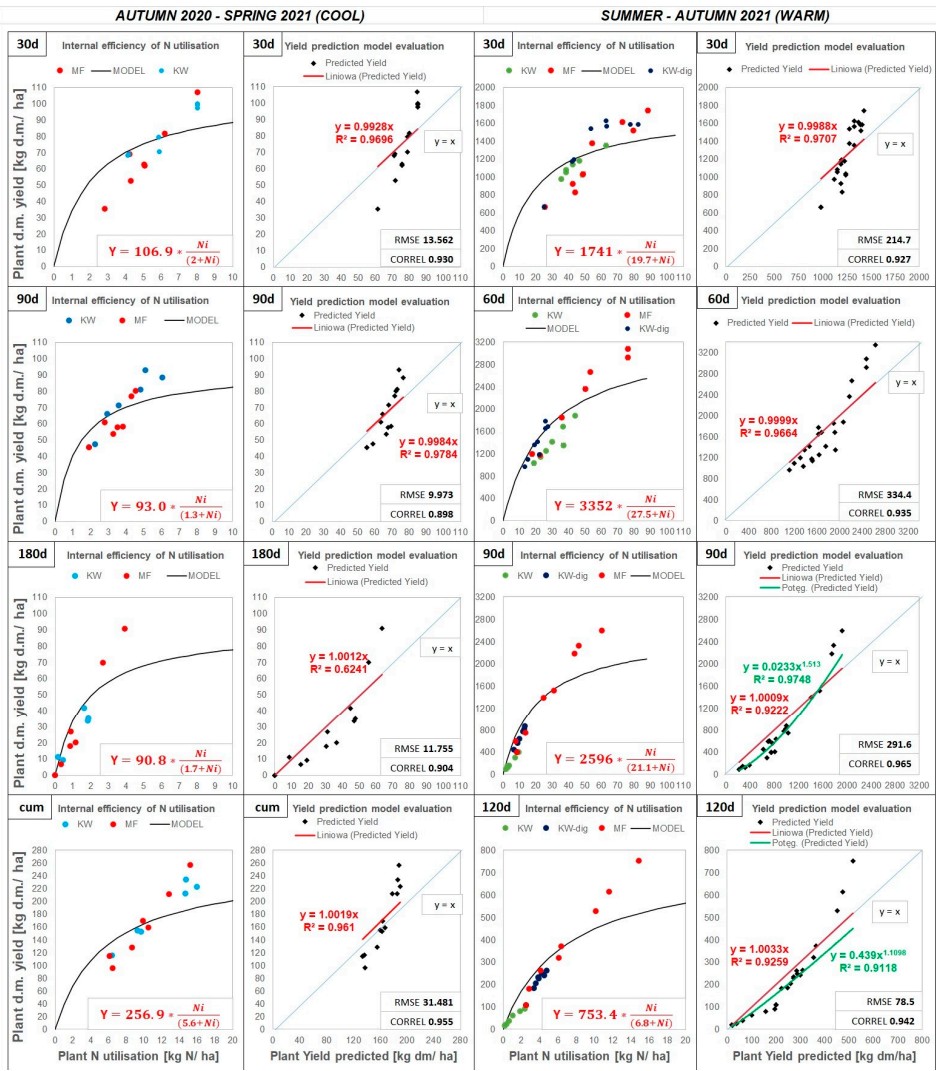

**Figure 5.** Internal efficiency of N utilization for all kitchen waste-based and mineral fertilizers for each harvest in the cool (**left**) and warm (**right**) seasons. Standard deviations included, insignificant if not visible. Red color equations describe the fitted Monod model's curves. (*) means a multiplication sign.

The coefficient of determination $R^2$, approaching high values (>0.92) for the slope of the function equal to 1, indicates the good fit of the most measured values to the Monod equation. An exception was observed for 180 days of growth in the cool season; $R^2$ was only 0.62. This indicates that N was the only growth limiting factor, especially at the beginning of growth, namely after 30–90 days in the cool season and 30–60 days in the warm season ($R^2$ reaching 0.97–0.98). The exceptionally high accuracy with the Monod model ($R^2 = 0.97$) was found for 90 days growth in the warm season, while slightly lower accuracy ($R^2 = 0.93$) was found for 120 days growth, where the internal efficiency curve has not reached the plateau. Not reaching the plateau indicates a still high nitrogen buffer capacity of the soil in the cooler months of the warm season (September–October) where applying high loads of fertiliser does not inhibit the growth of the plants. This could be related to the ability of plants to accumulate N by their roots as temperatures decrease in autumn for further supply of this stored N to the kernels in a more gradual and systematic way.

Figure 6 shows the applicability of the Monod models for the prediction of the dry matter yield as a function of N utilisation per ha. For the last two scenarios in the warm season (90 and 120 days), the power model was tested. The comparison is expressed by three precision parameters: square root ($R^2$), root mean square error (RMSE), and correlation coefficient (CORREL). A subjective ranking is expressed by points given for the quality of each parameter. The best parameters are coloured green, while the worst are coloured red. Taking into account all three precision parameters, the best Monod model fits were found for 30 and 90 days (cool season) and for 30 and 60 days (warm season). This indicates that in the first half of growth, nitrogen is the most relevant growth-stimulating parameter. A similar modelling approach was used in other studies [32,33].

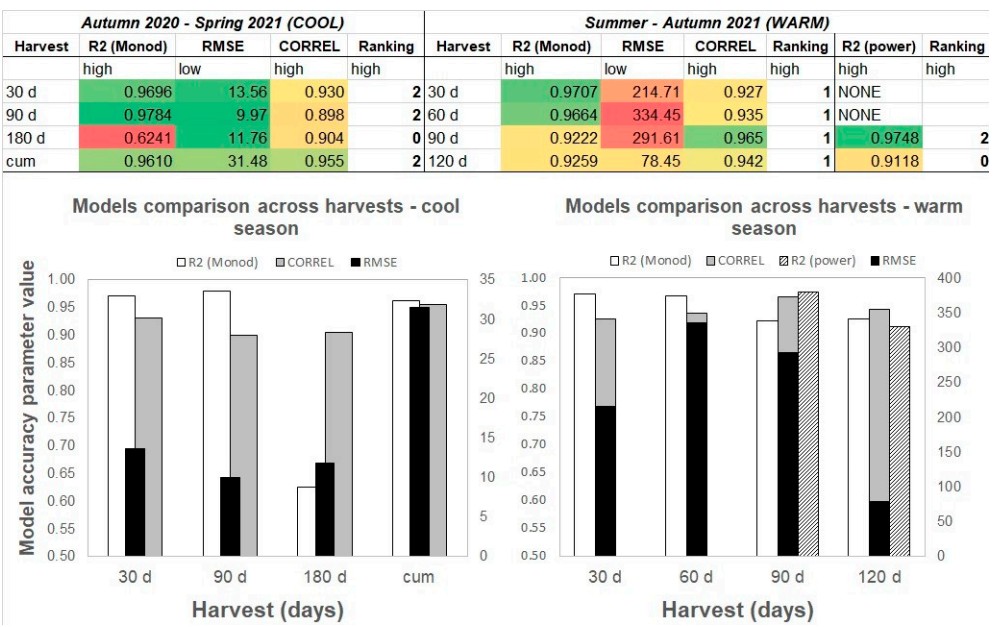

| Autumn 2020 - Spring 2021 (COOL) | | | | Summer - Autumn 2021 (WARM) | | | | | |
|---|---|---|---|---|---|---|---|---|---|
| Harvest | R2 (Monod) | RMSE | CORREL | Ranking | Harvest | R2 (Monod) | RMSE | CORREL | Ranking | R2 (power) | Ranking |
| | high | low | high | high | | high | low | high | high | high | high |
| 30 d | 0.9696 | 13.56 | 0.930 | 2 | 30 d | 0.9707 | 214.71 | 0.927 | 1 | NONE | |
| 90 d | 0.9784 | 9.97 | 0.898 | 2 | 60 d | 0.9664 | 334.45 | 0.935 | 1 | NONE | |
| 180 d | 0.6241 | 11.76 | 0.904 | 0 | 90 d | 0.9222 | 291.61 | 0.965 | 1 | 0.9748 | 2 |
| cum | 0.9610 | 31.48 | 0.955 | 2 | 120 d | 0.9259 | 78.45 | 0.942 | 1 | 0.9118 | 0 |

**Figure 6.** Comparison of the accuracy of the Monod kinetics model. RMSE values are displayed on a secondary axis. Numbers in cells are coloured proportionally to the models accuracy: the greener the model more accurate, the more red the model less accurate. Column called "Ranking" denotes individual ranking scores.

## 5. Discussion

### 5.1. Absolute Agronomic Effectiveness

Absolute agronomic effectiveness (AAE) is expressed by the slope of the best-fit linear regression model for the response of plant growth to fertiliser dosage. It could be expressed by the relationship between (1) N use or (2) dry-matter yield as a function of fertilizer dosage. In other words, N-based AAE describes how much nitrogen (in %) is taken up in

relation to nitrogen input (introduced and supplied with the fertilizer) across the increasing dosages. Calculating AAE is a common way to assess fertilizer performance throughout the literature and has been found in other studies [38,39].

### 5.1.1. Cool Season

N Utilization.

Because the ryegrass growth response to MF mostly did not represent a linear trend (very poor regression only for 90 days growth), AAE was only red from the plots in Figure 3 for KW. They are 0.0164 (30 days), 0.0139 (90 days), 0.0081 (180 days), and 0.0384 (cumulatively).

### 5.1.2. Warm Season

N Utilization.

Taking into account weak-to-moderate regressions for N use responses from Figure 3, the AAE of tested materials for four subsequent harvests were as follows: 0.1447, 0.2227, 0.1364, and 0.0326 for MF (showing the best N utilisation after 60 days); 0.1187, 0.0295, 0.0171, and 0.005 for KW–dig (the best N utilization after 30 days) and 0.0621, 0.0666, 0.0112, and 0.0032 for KW (no single best N utilization and significant drop in the last 2 months). This again shows that the anaerobic digestion of KW and its application on land generates AE comparable to AE of MF after 30 days and around 50–100% better AE than for undigested KW.

### *5.2. Relative Agronomic Effectiveness*

Figure 7 summarizes all previously presented data by showing the calculated relative agronomic effectiveness for the fertilizer materials, where it was possible to calculate AAE. RAE is actually the AAE related to the AAE of the reference mineral fertilizer. The RAE based on the utilisation of N (RAE(N)) was calculated and presented for each subsequent harvest and also as a total value after adding all the harvest data (total N). However, the d.m. yield-based RAE(Y) was calculated only as a total value (Total Y). N-utilization-based RAE(N) better characterizes the fertilizer material as it also contains the dry-matter yield when calculating the total N-utilisation per area. In general, KW works better than MF in the cool season, but RAE was only shown for 90 days (386.1%) due to very poor linear regression of the corresponding MF response as a reference. Therefore, it was not possible to calculate the RAE for KW in the cool season for each harvest, but it was nevertheless always higher than the RAE of MF. Under warm conditions, for the next 4 months, KW showed RAE in decreasing values: 43%, 29%, 8%, and 9%. KW-dig mostly demonstrated higher values: 82%, 13%, 13%, and 15% for subsequent harvests. In total, the RAE of KW–dig was 32% versus 27% for KW (N-based) and even 36% versus 21% (yield-based).

In spite of an observed improved performance of KW during the winter months, the presented approach is not only applicable to regions with a cool climate. KW, especially when digested, performed quite well under warm conditions (up to 82% of RAE at the 2nd growth month). This information could be used by urbanists, city planners, and green-area managers when using such prepared fertilizers in their monthly schedule while maintaining the urban green areas.

Of course, increasing mineral N, P, and K, and supplementing them with additional micronutrients, would surely improve the performance of the material, but this is not the point. We try to utilize the nature by bringing back the nutrients from the organic waste material to the soil, with the least-possible treatment avoiding further chemical supplementation, especially when the KW-based fertilizer meets the minimal requirements for the organic fertilizers according to the Polish "Act on Fertilizers and Fertilization." These requirements are: Total Organic Solids 30%, Total N 0.3%, Total $P_2O_5$ 0.2%, Total $K_2O$ 0.2%, heavy metals within limits, no parasite eggs (*Ascaris* sp., *Trichuris* sp., *Toxocara* sp.), and *Salmonella*. KW properties are way above these requirements: Total Organic

Solids 93.5%, Total N 3.4%, Total $P_2O_5$ 0.34%, and Total $K_2O$ 1.02%. So, there is no need to supplement with additional N, P, and K.

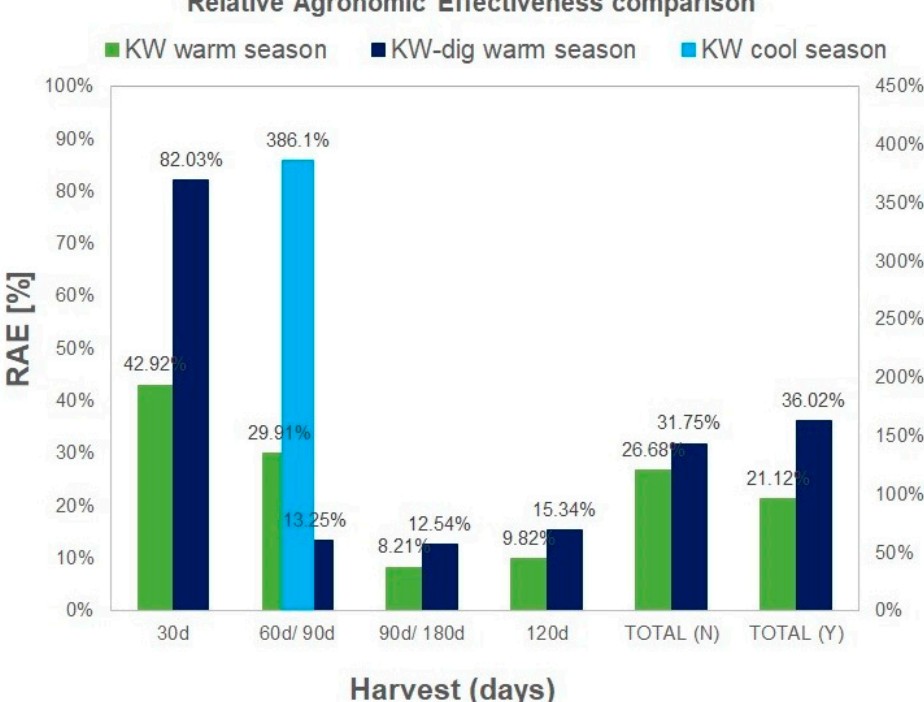

**Figure 7.** Comparison of RAE for the kitchen waste for each of the three harvests (cool season: 30, 90, 180 d) or four harvests (warm season: 30, 60, 90, 120 d) and the total N-based and yield-based after 120 days of growth. Values for KW (cool season) are displayed on a secondary axis. Exact values of RAE in % are displayed close to the bars.

## 6. Conclusions and Perspectives

Source-separated food waste at the household level (kitchen waste) is a clean and abundant source of nutrients and carbon that could constitute a feedstock for organic fertilizer manufacturing. In this paper, pellets from both raw kitchen waste (incubated with EM) and anaerobically digested kitchen waste have been verified as organic fertilizers in both the cool and warm seasons via glasshouse studies.

In general, kitchen waste performed better in the cool season than in the warm season, providing 30–100% higher growth and N uptake in higher dosages (>170 kg N/ha) than mineral fertilizer, which resulted in over-fertilisation for MF and the inhibition of growth in these applications. Under warm conditions, the relative agronomic effectiveness of kitchen waste (KW) decreased over time from 43% to 9% for undigested material (EM-incubated) and from 82% to 13% for anaerobically digested material (KW–dig). This makes anaerobic digestion a desirable step for improving energy extraction and nutrient availability prior to fertilizer manufacturing.

The best Monod model fits ($R^2 > 0.966$) were found for 30 and 90 days (cool season) and for 30 and 60 days (warm season). This indicates that in the first half of growth, nitrogen is the most relevant growth-stimulating parameter. The residual soil parameters were best after treatment of the undigested kitchen waste (KW). Namely, pH remained stable, salinity decreased, and soil N remained mostly unchanged (during both seasons). This is unlike the application of mineral fertilizer where pH slightly decreased (mainly in the cool season), salinity increased (both seasons), and some N was lost through probable denitrification and $N_2O$ emissions (mainly in the warm season).

The digested kitchen waste (KW–dig) increased soil salinity, slightly decreased pH, and kept residual N at the initial levels. The study shows that (1) Organic kitchen waste applied to soil performs better in the cool season; and (2) The treatment of kitchen waste

with anaerobic digestion increases the effectiveness of the fertilizer in relation to their incubation with effective microbes. More studies on varying kitchen waste morphology and pre-treatment methods should further address this issue.

The research was performed with the following limitations: (1) It was carried out in the glasshouse; (2) Small pots were used; (3) It was held under semi-controlled meteorological circumstances; (4) A limited number of harvests were held during the study; (5) One crop and one soil type were used; (6) An artificially prepared kitchen waste model was used; (7) An anaerobic digestion performed under laboratory conditions was included; and (9) The setting involved lab-scale bioreactors. These limitations imply that the future works should back up these findings by testing different source-separated kitchen waste (by households), possibly using long-term field trials and having the digestates from the commercially operating biogas plant.

The use of kitchen waste as organic fertilizers for crop growth is a promising area of research with several future perspectives. While the results of this study are promising, they were conducted in a glasshouse environment. There is a need to explore the feasibility of scaling up the application of kitchen waste fractions as fertilizers in field conditions. **Large-scale field trials** could provide insights into the potential of these fertilizers in improving crop yields and the long-term effects on soil health. Field trials could also provide data on the optimal application rates and timing for these fertilizers to achieve maximum benefits.

Furthermore, the challenges associated with scaling up the production and the application of these fertilizers need to be addressed. These challenges may include issues such as **transportation, storage, and application methods**. It is essential to develop cost-effective and practical solutions to these challenges to ensure the successful adoption of kitchen waste as organic fertilizers in agriculture.

In addition, this study focused on ryegrass growth, but it would be interesting to investigate the effectiveness of kitchen waste fractions as organic fertilizers for **other crops** as well. Different crops have varying nutrient requirements, and studying the impact of these fertilizers on a wider range of crops could help identify their potential for use in diverse agricultural systems. Studying the impact of these fertilizers on a wider range of crops could help identify their potential for use in **diverse agricultural systems**. This could include crops that are commonly grown in different regions, crops with high nutrient requirements, and crops that are susceptible to diseases and pests. Furthermore, investigating the impact of these fertilizers on a wider range of crops could also provide insights into their potential for use in-**crop-rotation** systems. Crop rotation is an essential practice in sustainable agriculture as it helps to maintain soil fertility, prevent diseases and pests, and reduce the use of synthetic fertilizers and pesticides.

Furthermore, a comprehensive **environmental impact assessment** is needed to evaluate the broader implications of their use in agriculture. This could include assessing the impact of these fertilizers on soil health, water quality, and greenhouse gas emissions. Source-separated kitchen waste (if not sorted properly) may contain heavy metals or other contaminants (i.e., microplastics) that could potentially be harmful to soil and water resources if not managed properly. Therefore, it is crucial to investigate the **potential risks** associated with their use and develop appropriate management practices to mitigate these risks. Moreover, the production and application of these fertilizers could potentially contribute to greenhouse gas emissions, depending on the methods used. Therefore, it is important to assess the **carbon footprint** associated with their production and use and develop strategies to minimize their impact on the environment.

There is also a need to explore the commercial potential of using kitchen waste fractions as organic fertilizers. This could involve developing cost-effective production methods for these fertilizers and exploring potential markets for their sale. Finally, future research could focus on **optimizing the production** processes of these fractions to maximize their effectiveness as fertilizers while minimizing any negative environmental impacts.

**Author Contributions:** K.K.: Conceptualization, Data curation, Formal analysis, Funding acquisition, Project administration, Investigation, Methodology, Resources, Supervision, Validation, Visualization, Writing—original draft. I.K.: Data curation, Investigation, Resources, Validation, Writing—review & editing. L.Ś.: Investigation, Validation, Writing—review & editing. A.C.: Project administration, Supervision, Validation, Writing—review & editing. K.C.: Validation, Writing—review & editing. S.S.: Validation, Writing—review & editing. All authors have read and agreed to the published version of the manuscript.

**Funding:** This research was co-funded under the project No. STHB.02.02.00-22-0131/17 co-financed by the European Regional Development Fund, entitled: WasteMan—Integrated Sustainable Waste Management Systems decreasing pollution discharges in the South Baltic area (INTERREG South Baltic Programme 2014–2021), webpage: https://www.imp.gda.pl/wasteman/, accessed on 22 March 2023. Lead Partner: The Institute of Fluid-Flow Machinery Polish Academy of Sciences. This research was also co-funded by Miniserstwo Nauki I Szkolnictwa Wyższego (Ministry of Science and Higher Education of Poland) within the "Projekty Międzynarodowe Współfinansowane" (International Projects Co-financed) Programme.

**Institutional Review Board Statement:** Not applicable.

**Informed Consent Statement:** Not applicable.

**Data Availability Statement:** Not applicable.

**Acknowledgments:** The authors would like to express their gratitude to the lab technician Sabina Szymańska for her contribution to harvest assistance, sample preparation, and analyses (dry-matter and N contents) and to Adrian Woźniak and Aleksandra Woźniak for the assistance with watering the plants in Glasshouse 2.

**Conflicts of Interest:** The authors declare that they have no known competing financial interests or personal relationships that could have appeared to influence the work reported in this paper. The funders had no role in the design of the study; in the collection, analyses, or interpretation of data; in the writing of the manuscript; or in the decision to publish the results.

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
