# Peer review of "Evaluation of Kitchen Waste Recycling as Organic N-Fertiliser for Sustainable Agriculture under Cool and Warm Seasons"

_sustainability, doi:10.3390/su15107997_

Round 1

Reviewer 1 Report

A literature review is completely missing. It would be expected that the state of the art on waste recycling.

Conclusion

Can the authors critically reflect on their work and add a limitation section--> Research approach and sampling. 

Can the authors make suggestions for future research

Author Response

REVIEWER 1:

  1. A literature review is completely missing. It would be expected that the state of the art on waste recycling.

Answer: The introduction has been modified. However we disagree with shortening it due to the fact of being closely related to inter-disciplinary character of the paper: agricultural application of kitchen waste but also utilisation of residues from energy-recovery processes. In other words, shortening only to “waste recycling” aspect would make the introduction poorer and not linked to the research results itself.

  1. Conclusion

Answer: The Conclusion has been extended by adding limitations and directions (perspectives) for the future research.

  1. Can the authors critically reflect on their work and add a limitation section--> Research approach and sampling.

Answer: An appropriate paragraph was added to the “Materials and Methods”

“2.4 Limitation

The research was performed with the following limitations: (1) carried out in the glasshouse, (2) using small-diameter pots, (3) under semi-controlled meteorological circumstances, (4) for the limited amount of harvests, (5) for one crop and (6) one soil type and (7) using model (artificially prepared) kitchen waste including (8) anaerobic digestion per-formed under laboratory conditions in the (9) lab-scale bioreactors.”

  1. Can the authors make suggestions for future research

Answer: An extensive paragraph on perspectives has been added to the Conclusions section. “Conclusions” section has changed into “Conclusions and Perspectives”.

Reviewer 2 Report

Comments on “Evaluation of kitchen waste recycling as organic N-fertiliser for sustainable agriculture: Effectiveness and biological modelling under cool and warm season” 

In this manuscript author developed a safe and environment friendly fertiliser for sustainable agriculture from kitchen waste by treating with anaerobic Effective Micro- organisms (EM), and anaerobically digested (AD) KW. They were applied in two separate glasshouse experiments: one under cool and one under warm season and found that in cool season the fertilizer is more effective than warm season. Author has provided plenty of experimental evidence to support their observation, however, I believe that the current version of the manuscript still needs improvement before it can be published. Author can address following comments in their revised manuscript.

1.      The author made an effort to include all the background information in the introduction. But, I feel that the introduction is simply too long. It should be streamlined to only cover the main elements and the work's objectives in order to make it easier to read.

2.      Since kitchen waste fertilizers are only useful during the cool months, is the model applicable only to countries with cool climates?

3.      Can the kitchen waste fertilizer's component ratio and component formula be changed to boost its efficacy during the warm months as well?

Author Response

REVIEWER 2:

Comments on “Evaluation of kitchen waste recycling as organic N-fertiliser for sustainable agriculture: Effectiveness and biological modelling under cool and warm season”

In this manuscript author developed a safe and environment friendly fertiliser for sustainable agriculture from kitchen waste by treating with anaerobic Effective Micro-organisms (EM), and anaerobically digested (AD) KW. They were applied in two separate glasshouse experiments: one under cool and one under warm season and found that in cool season the fertilizer is more effective than warm season. Author has provided plenty of experimental evidence to support their observation, however, I believe that the current version of the manuscript still needs improvement before it can be published. Author can address following comments in their revised manuscript.

  1. The author made an effort to include all the background information in the introduction. But, I feel that the introduction is simply too long. It should be streamlined to only cover the main elements and the work's objectives in order to make it easier to read.

Answer: The introduction has been modified. However we disagree with shortening it due to the fact of being closely related to inter-disciplinary character of the paper: agricultural application of kitchen waste but also utilisation of residues from energy-recovery processes. In other words, shortening only to “waste recycling” aspect would make the introduction poorer and not linked to the research results itself.

  1. Since kitchen waste fertilizers are only useful during the cool months, is the model applicable only to countries with cool climates?

Answer: Authors do not agree with the reviewer stating that KW-based fertilisers are only “useful” during the cool months. We proved that they perform better in these months but they are still useful in warm months, expressing up to 82% of RAE (especially for digested ones). An appropriate sentence was added to the discussion:

“In spite of observed better performance of KW during winter months, the presented approach is not only applicable to regions with cool climate. KW, especially digested one performed quite well under warm conditions (up to 82% of RAE at the 2nd growth month). This information could be used by the urbanists, city planners and green area managers when using such prepared fertilisers in their monthly schedule when maintaining the urban green areas.”

  1. Can the kitchen waste fertilizer's component ratio and component formula be changed to boost its efficacy during the warm months as well?

Answer: Of course increasing mineral N, P and K and supplementing with additional micronutrients would surely improve the performance of the material, but this is not the point. We try to utilise the nature by bringing back the nutrients from the organic waste material to the soil, with the least possible treatment avoiding further chemical supplementation. However the KW-based fertiliser meets the minimal requirements for the organic fertilisers according to the Polish “Act on fertilisers and fertilisation”. They are: Total Organic Solids 30%, Total N 0.3%, Total P2O5 0.2%, Total K2O 0.2%, heavy metals within limits, no parasite eggs (Ascaris sp., Trichuris sp., Toxocara sp.) and Salmonella. KW properties are way above these requirements: Total Organic Solids 93.5%, Total N 3.4%, Total P2O5 0.34%, Total K2O 1.02%, so no need to supplement with additional N, P and K. An appropriate paragraph was added to the discussion.

Reviewer 3 Report

The article’s topic is incredibly intriguing from both a content and a visual standpoint. The authors' thorough preparation for the analysed subject is reflected in their use of a large body of relevant literature. Nevertheless, in order to enhance the quality of the manuscript, I suggest that the authors follow the steps that are listed below:

Point 1: The title is quite lengthy; the authors ought to make use of headings that are concise and pertinent.

Point 2: The authors didn't consistently employ the correct sustainability citation style throughout the manuscript.

Point 3: From Line 46 to Line 65 and 169 to 185 The authors have lifted large chunks of text directly from WP4 Reports without attributing the original work or providing a citation.  Available online https://www.imp.gda.pl/wasteman/artykuly/WasteAsFertilisers.pdf

In the realm of science, making duplicate submissions of any part of research that are identical to one another is seen as unethical. These paragraphs are recommended to be rewritten by the authors.

Point 4: In the section titled "Materials and Methods," the authors describe how they carried out their investigation by performing pot experiments in two greenhouses. However, the general audience won't be able to comprehend the image without a proper visualisation of all steps, such as model preparation, kitchen waste conversion, fertiliser production, soil preparation, and fertiliser application.

Point 5: In the “Results” section, authors are expected to describe their own findings, but in the case of COOL SEASON (DAYS 30-180), lines 346–378 and 497-521, the authors have plagiarised the entire experiment from WP4 Reports. Available online https://www.imp.gda.pl/wasteman/artykuly/WasteAsFertilisers.pdf

The authors are requested to rewrite these findings to avoid "Self-plagiarism," which would be a violation of laws governing copyright.

Point 6: Adding limitations of the research to the conclusion is a good practice. 

Author Response

REVIEWER 3:

The article’s topic is incredibly intriguing from both a content and a visual standpoint. The authors' thorough preparation for the analysed subject is reflected in their use of a large body of relevant literature. Nevertheless, in order to enhance the quality of the manuscript, I suggest that the authors follow the steps that are listed below:

  1. The title is quite lengthy; the authors ought to make use of headings that are concise and pertinent.

Answer: The title has been shortened to: “Evaluation of kitchen waste recycling as organic N-fertiliser for sustainable agriculture under cool and warm seasons”

  1. The authors didn't consistently employ the correct sustainability citation style throughout the manuscript.

Answer: The citations have been corrected to fit the journal requirements.

  1. From Line 46 to Line 65 and 169 to 185 The authors have lifted large chunks of text directly from WP4 Reports without attributing the original work or providing a citation. Available online https://www.imp.gda.pl/wasteman/artykuly/WasteAsFertilisers.pdf

Answer: Yes, I am aware of that but since (1) the WASTEMAN Project report is an internal project document (not a peer-reviewed journal publication) and its first author is the same first author of this manuscript, I wouldn’t see it as the problem. But, of course, the text has been modified so it is not exactly the same. Lines 46-65 is the national/ EU statistics so it is hardly possible to write it differently than it is in the original as it is mostly numbers (facts). Lines 169-185 explain the motivation to perform the current study based on previous studies, and yes, this was mentioned in the WASTEMAN Project Report too as a starting point to our works. An appropriate citation was added under lines 46-65 and 169-185 and corresponding reference was added:

“Kuligowski, K., Cenian, A., Åšwierczek, L., Konkol, I. 2021. Municipal solid waste organic fraction (kitchen waste) management via urban green areas fertilization WP4 Report. Interreg Soutb Baltic Programme WASTEMAN Project Internal Report, available here: https://www.imp.gda.pl/wasteman/artykuly/WasteAsFertilisers.pdf”

  1. In the realm of science, making duplicate submissions of any part of research that are identical to one another is seen as unethical. These paragraphs are recommended to be rewritten by the authors.

Answer: Explaining: This is not a duplicate submission as WASTEMAN Project Report is an internal report, not being submitted to any other journals. It also only contains the cool season data and in the format that is not normally accepted by peer-reviewed journals. Moreover self-plagiarism does not exist, plagiarism is when an author X uses literally the same text of an author Y. In this case it is the same group of authors and we are just developing the work further. I have provided the citations in the text and left the paragraphs in slightly changed versions.

  1. In the section titled "Materials and Methods," the authors describe how they carried out their investigation by performing pot experiments in two greenhouses. However, the general audience won't be able to comprehend the image without a proper visualisation of all steps, such as model preparation, kitchen waste conversion, fertiliser production, soil preparation, and fertiliser application.

Answer: Firstly all details are explained in Sections 2.1 Fertilisers and 2.2 Soil and plants. This is a common way to present the methodology in soil science/ agricultural studies. If needed a proper citations were used too. Below I am emphasising the already existing (under Section 2.1) paragraph that contains part mentioned by the Reviewer steps. Remaining two (soil preparation and fertiliser application) are under Section 2.2 and Table 1 and 2:

“First, the model kitchen waste (Paulsrud et al., 2016) has been prepared, according to the following recipe: 25 g each of apple, lemon, roll, butter, sour cream, milk, cottage cheese, yoghurt, eggs, meat with bones, sausage, fish meat, potatoes, banana, tomato, lettuce, fruit juice, bun and 50 g of flowers and paper were ground to a particle size less than 5 mm and mixed well to obtain homogeneous mass (MODEL PREPARATION). The obtained basic substrate was then processed into two fertilisers. KW fertiliser was obtained by the basic substrate inoculation with commercially available (mainly anaerobic) effective microorganisms (Greenland Technologia EM LtD.). Inoculation was achieved by dispersing 1 ml of the bacterial product in 250 ml of deionized water, and mixed with 1 kg of sub-strate. The substrate was then collected in a sealed plastic container with a vent tube for two weeks (KITCHEN WASTE CONVERSION and FERTILISER PRODUCTION 1). After fermentation process, the substrate was partly dried to about 70% dry mass and formed into pellet-shaped granules, then dried to obtain the stable mass (ca. 95% d.m.). KW-dig fertiliser consisted of residues after methane digestion of KW. Mesophilic methane digestion was carried out in 2 L reactors for 30 days in accordance with the methodology described in (Konkol et al., 2018). The fermentation residue was centrifuged in a laboratory centrifuge (MPW 260RH) for 10 min at 5000 RPM without the use of coagulants. The prepared KW-dig fertiliser was stored at 4°C until used in the greenhouse tests (KITCHEN WASTE CONVERSION and FERTILISER PRODUCTION 2).

In order to follow the step-wise methodology, new sub-headings were introduced under “Materials and Methods” in Sections 2.1 and 2.2.

  1. Point 5: In the “Results” section, authors are expected to describe their own findings, but in the case of COOL SEASON (DAYS 30-180), lines 346–378 and 497-521, the authors have plagiarised the entire experiment from WP4 Reports. Available online https://www.imp.gda.pl/wasteman/artykuly/WasteAsFertilisers.pdf

Answer: Explaining: The WASTEMAN Project Report is an internal report, not being submitted to any other journals. It also only contains the cool season data and in the format that is not normally accepted by peer-reviewed journals. Moreover self-plagiarism does not exist, plagiarism is when an author X uses literally the same text of an author Y. In this case it is the same group of authors and we are just developing the work further. I have provided the citations in the text and left the 346-378/ 497-521 lines in modified versions to avoid the same wording.

  1. The authors are requested to rewrite these findings to avoid "Self-plagiarism," which would be a violation of laws governing copyright.

Answer: See above.

  1. Adding limitations of the research to the conclusion is a good practice.

Answer: The following paragraph was added to the Conclusions section:

“The research was performed with the following limitations: (1) carried out in the glasshouse (2) using small pots (3) under semi-controlled meteorological circumstances, (4) for limited amount of harvests, (5) for one crop and (6) one soil type and (7) using model (artificially prepared) kitchen waste including (8) anaerobic digestion per-formed under laboratory conditions in the (9) lab-scale bioreactors. These limitations imply that the future works should back up these findings by testing different source-separated kitchen waste (by households), possibly using long-term field trials and having the digestates from the commercially operating biogas plant.”

Reviewer 4 Report

The manuscript by Ksawery Kuligowski et al reported the investigation of the temperature effect on kitchen waste treated by effective microorganisms and anaerobically digested. The manuscript is well organized and the result is sufficiently supported by the experiments. This work could serve as the guidance for future fertilizer improvement. I would recommend minor revision for this work.

Minor issues:

1.      Avoid directly use abbreviated terms in the abstract, NPK fertilizer.

2.      Check the format of manuscript, line 660: R2 should be R2, line 667 N2O should be N2O.

3.      Re-compose the figure 7, the percent number is overshadowed by the column.

4.      Check the references, reference 2 missed the DOI.

Author Response

REVIEWER 4

The manuscript by Ksawery Kuligowski et al reported the investigation of the temperature effect on kitchen waste treated by effective microorganisms and anaerobically digested. The manuscript is well organized and the result is sufficiently supported by the experiments. This work could serve as the guidance for future fertilizer improvement. I would recommend minor revision for this work.

Minor issues:

  1. Avoid directly use abbreviated terms in the abstract, NPK fertilizer.

Answer: Abbreviations removed from the Abstract and full names were introduced instead.

  1. Check the format of manuscript, line 660: R2 should be R2, line 667 N2O should be N2O.

Answer: Formats corrected.

  1. Re-compose the figure 7, the percent number is overshadowed by the column.

Answer: This is a minor problem. We don’t want to change the colour of the KW-dig data (dark blue) as this colour is reserved for this material on all Figs. 1-7. I have included the explanation in the Fig.  7 caption that should do the work: “Exact values of RAE in % are displayed close to the bars.”

  1. Check the references, reference 2 missed the DOI.

Answer: DOI for such an old publication not found.

Round 2

Reviewer 3 Report

In the revised version of the manuscript, the author has made significant advancements and answered all of the questions that were raised. Nevertheless, I would like to make a few minor suggestions for the improvement of this manuscript:

Point 1: It is evident that the authors included imitations twice. First, included 2.4. Limitations (lines 324 to 329) in Materials and Methods, which are inappropriate.

2.4. Limitation

The research was performed with the following limitations: (1) carried out in the glass-house, (2) using small-diameter pots, (3) under semi-controlled meteorological circum- stances, (4) for the limited amount of harvests, (5) for one crop and (6) one soil type and (7) using model (artificially prepared) kitchen waste including (8) anaerobic digestion performed under laboratory conditions in the (9) lab-scale bioreactors.

The identical limitation appears multiple times throughout paragraphs (line no. 694 to 698) of the conclusion. However, mentioning these limitations in the concluding part would be appropriate.

Please remove the “2.4 Limitation” paragraph from the Materials and Methods section.

Point 2: The multiple titles are identified under the following figures:

Line no. 349 please delete "Fig. 1 HERE"

Line no. 426 please delete "Fig. 2 HERE"

Line no. 470 please delete "Fig. 3 HERE"

Line no. 505 please delete "Fig. 4 HERE"

Line no. 549 please delete "Fig. 5 HERE"

Line no. 584 please delete "Fig. 6 HERE"

Line no. 629 please delete "Fig. 7 HERE"

Point 3: Please ensure that lines 609 and 616, labelled "N utilisation," are formatted with the appropriate subheading style.

Author Response

In the revised version of the manuscript, the author has made significant advancements and answered all of the questions that were raised. Nevertheless, I would like to make a few minor suggestions for the improvement of this manuscript:

Point 1: It is evident that the authors included imitations twice. First, included 2.4. Limitations (lines 324 to 329) in Materials and Methods, which are inappropriate.

2.4. Limitation

The research was performed with the following limitations: (1) carried out in the glass-house, (2) using small-diameter pots, (3) under semi-controlled meteorological circum- stances, (4) for the limited amount of harvests, (5) for one crop and (6) one soil type and (7) using model (artificially prepared) kitchen waste including (8) anaerobic digestion performed under laboratory conditions in the (9) lab-scale bioreactors.

The identical limitation appears multiple times throughout paragraphs (line no. 694 to 698) of the conclusion. However, mentioning these limitations in the concluding part would be appropriate.

Please remove the “2.4 Limitation” paragraph from the Materials and Methods section.

ANSWER: Done

Point 2: The multiple titles are identified under the following figures:

Line no. 349 please delete "Fig. 1 HERE"

Line no. 426 please delete "Fig. 2 HERE"

Line no. 470 please delete "Fig. 3 HERE"

Line no. 505 please delete "Fig. 4 HERE"

Line no. 549 please delete "Fig. 5 HERE"

Line no. 584 please delete "Fig. 6 HERE"

Line no. 629 please delete "Fig. 7 HERE"

ANSWER: Done

Point 3: Please ensure that lines 609 and 616, labelled "N utilisation," are formatted with the appropriate subheading style.

ANSWER: Done